# CODA: Generalizing to Open and Unseen Domains with Compaction and Disambiguation

**Chaoqi Chen**[1]* **Luyao Tang**[2]* **Yue Huang**[2] **Xiaoguang Han**[3]† **Yizhou Yu**[1]†

[1]The University of Hong Kong [2]Xiamen University
[3]The Chinese University of Hong Kong (Shenzhen)
cqchen1994@gmail.com, lytang@stu.xmu.edu.cn, yhuang2010@xmu.edu.cn
hanxiaoguang@cuhk.edu.cn, yizhouy@acm.org

## Abstract

The generalization capability of machine learning systems degenerates notably when the test distribution drifts from the training distribution. Recently, Domain Generalization (DG) has been gaining momentum in enabling machine learning models to generalize to unseen domains. However, most DG methods assume that training and test data share an identical label space, ignoring the potential unseen categories in many real-world applications. In this paper, we delve into a more general but difficult problem termed Open Test-Time DG (OTDG), where both *domain shift* and *open class* may occur on the unseen test data. We propose Compaction and Disambiguation (CODA), a novel two-stage framework for learning compact representations and adapting to open classes in the wild. To meaningfully regularize the model's decision boundary, CODA introduces virtual unknown classes and optimizes a new training objective to insert unknowns into the latent space by compacting the embedding space of source known classes. To adapt target samples to the source model, we then disambiguate the decision boundaries between known and unknown classes with a test-time training objective, mitigating the adaptivity gap and catastrophic forgetting challenges. Experiments reveal that CODA can significantly outperform the previous best method on standard DG datasets and harmonize the classification accuracy between known and unknown classes.

## 1 Introduction

The ability to generalize to unseen environments is considered a key signature of human intelligence [12]. While deep neural networks have achieved great success in many machine learning problems, they are brittle to distribution shifts between training and test domains, which often occur in real-world applications. For example, when deploying object recognition systems in autonomous vehicles, the ever-changing weather conditions (*e.g.* fog, rain, and snow) may deteriorate the performance and raise concerns about their reliability over time. This motivates a challenging scenario named Domain Generalization (DG) [68, 84], which extrapolates learning machines to related yet previously unseen test domains by identifying the common factors from available source data.

From the perspective of representation learning, the mainstream paradigm for DG includes invariant risk minimization [2, 1, 89], domain alignment [31], feature disentanglement [45, 36, 78], meta-learning [29, 30], and augmentation-based invariant prediction [66, 74, 88]. In spite of the significant progress in DG, the adaptivity gap [14] between source and target domains naturally exists and emerges as an inevitable challenge. Therefore, some prior efforts [23, 24, 72, 11] strive to adapt

---

*First two authors contributed equally.
†Corresponding authors.

37th Conference on Neural Information Processing Systems (NeurIPS 2023).

the source-trained model through the lens of test-time adaptation [34], which uses unlabeled target samples in an online manner. While these methods alleviate the adaptivity gap to some extent, there is still no guarantee for any specific domain, especially if the domain divergence is large [3]. Moreover, most existing DG methods are designed with the assumption that the label space in the source and target domain are identical, which is too restrictive to be satisfied in practice.

To enable machine learning systems to be resilient to an open world, we aim at a more practical yet under-explored problem named Open Test-time Domain Generalization (OTDG), where both *domain shift* and *open class* occur in the unseen target data. The primary challenge of the proposed problem lies in addressing two critical aspects: (1) *model generalization* with an incomplete training label space, and (2) *online model adaptation* with asymmetric test label space.

Motivated by this, we propose a novel framework for OTDG, termed Compaction and Disambiguation (CODA). Our key idea is to enforce constraints on the decision boundaries during training using labeled source data and refine them during test time using unlabeled target data. To regularize the model's decision boundary and make the model expandable, CODA introduces a set of virtual unknown classes and optimizes a novel training objective in conjunction with the standard cross-entropy loss. The response to known- and unknown-class logits will be activated for both real and synthesized samples. This process embeds unknowns into the latent space by compacting the embedding space of known source classes, thereby reserving sufficient latent space for target unknown classes. To disambiguate the decision boundaries between known and unknown classes, we propose a novel prototype-based test-time adaptation pipeline. Specifically, the test-time classification should be subjected to three constraints: (1) consistency between the predicted class distributions and the estimated class conditionals, (2) class-wise sample reliability for ensuring the quality of target pseudo labels, and (3) semantic consistency between source-trained and present model predictions.

The main contributions are summarized as follows:

- We propose CODA, a simple and effective DG framework to mitigate domain shift and identify open classes in unseen test environments.

- We introduce a virtual unknown optimization process to make the model expandable for open classes, and a test-time training objective to match the real test data to corresponding known and unknown class patterns.

- We conduct extensive experiments and demonstrate that CODA outperforms previous methods on a series of DG benchmarks.

## 2  Preliminaries

**Problem setup.** Let us formally define the OTDG problem. We have access to a source domain $\mathcal{D}_s = \{(\mathbf{x}_s^i, y_s^i)\}_{i=1}^{n_s}$ of $n_s$ labeled data points and multiple unseen target domains $\mathcal{D}_t = \{(\mathbf{x}_t^j)\}_{j=1}^{n_t}$ of $n_t$ unlabeled data points. Let $\mathcal{C}_s$ and $\mathcal{C}_t$ be the source and target class sets, respectively. In OTDG, we have $\mathcal{C}_s \subset \mathcal{C}_t$ and $\mathcal{C}_t^u = \mathcal{C}_t \setminus \mathcal{C}_s$ is referred to as *unknown* classes. Assume that $\mathcal{X}$ is the input space, $\mathcal{Z}$ is the latent space, and $\mathcal{Y}$ is the output space. The predictor $f = h \circ g$ is comprised of a featurizer $g : \mathcal{X} \mapsto \mathcal{Z}$ that learns to extract embedding features, and a classifier $h : \mathcal{Z} \mapsto \mathcal{Y}$ that makes predictions based on the extracted features. The goal of OTDG is to find a predictor $f : \mathcal{X} \mapsto \mathcal{Y}$ that generalizes well to all unseen target domains. Although the labeling function $h_t$ is unknown, we assume that we have access to unlabeled instances from $\mathcal{D}_t$ at test time. In addition, we present the comparison of different problem settings in Table 1.

**Unknown-aware training.** To solve OTDG, a simple baseline is to train a $(|\mathcal{C}_s|+1)$-way classifier [90, 77, 9], where the additional dimension is introduced to identify the unknown. Formally, we define the standard cross-entropy loss as:

$$\mathcal{L}_{\text{CE}}(f(\mathbf{x}), y) = -\log \frac{\exp(f_k(\mathbf{x}))}{\sum_{c \in |\mathcal{C}_s|+1} \exp(f_c(\mathbf{x}))}, \tag{1}$$

where $f(\mathbf{x}) \in \mathbb{R}^{|\mathcal{C}_s|+1}$ denotes the network's logit and $f_k(\mathbf{x})$ is the $k$-th element of $f(\mathbf{x})$ corresponding to the ground-truth label $y$. As shown in Figure 1 (a), however, such optimization fails to activate the network's response to unknown classes. In this work, we first introduce a simple yet very effective baseline [9] for OTDG. Their key idea is to directly activate the unknown's logit by optimizing its

Table 1: Comparison of related machine learning problems. DA refers to Domain Adaptation and OOD stands for Out-of-Distribution. 'One-pass' indicates that target domain data only passes the network once during the whole process including the training and testing phases.

| Problem Setting | Training | | Test-time | | | | One-pass |
|---|---|---|---|---|---|---|---|
| | Training Data | Training Loss | Testing Loss | Domain shift | Open class | Adaptivity gap | |
| Open-Set DA | $\mathbf{x}_s, y_s, \mathbf{x}_t$ | $\mathcal{L}(\mathbf{x}_s, y_s) + \mathcal{L}(\mathbf{x}_s, \mathbf{x}_t)$ | – | ✓ | ✓ | ✓ | ✗ |
| Source-Free DA | $\mathbf{x}_t$ | $\mathcal{L}(\mathbf{x}_t)$ | – | ✓ | ✗ | ✓ | ✗ |
| OOD Detection | $\mathbf{x}_s, y_s$ | $\mathcal{L}(\mathbf{x}_s, y_s)$ | – | ✗ | ✓ | ✗ | ✓ |
| Test-Time Adaptation | $\mathbf{x}_s, y_s$ | $\mathcal{L}(\mathbf{x}_s, y_s)$ | $\mathcal{L}(\mathbf{x}_t)$ | ✓ | ✗ | ✓ | ✓ |
| Test-Time DG | $\mathbf{x}_s, y_s$ | $\mathcal{L}(\mathbf{x}_s, y_s)$ | $\mathcal{L}(\mathbf{x}_t)$ | ✓ | ✗ | ✓ | ✓ |
| Open-Set DG | $\mathbf{x}_s, y_s$ | $\mathcal{L}(\mathbf{x}_s, y_s)$ | – | ✓ | ✓ | ✗ | ✓ |
| OTDG (Ours) | $\mathbf{x}_s, y_s$ | $\mathcal{L}(\mathbf{x}_s, y_s)$ | $\mathcal{L}(\mathbf{x}_t)$ | ✓ | ✓ | ✓ | ✓ |

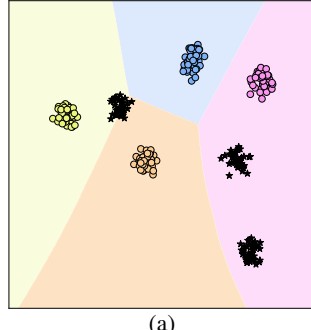 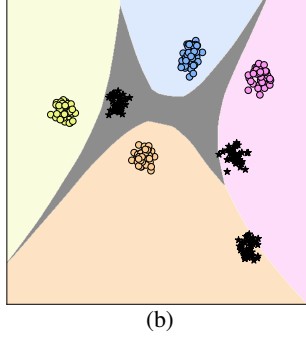 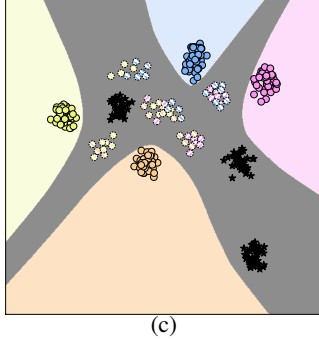

(a)            (b)            (c)

Figure 1: The decision boundaries learned by different unknown-aware training objectives: **(a)** Eq. 1, **(b)** Eq. 2, and **(c)** Eq. 3 + Eq. 4. These toy examples are generated by scikit-learn toolkit. Yellow, blue, orange, and pink points represent the known-class samples, while black points are unknown-class samples. Different clusters of back points stand for different unknown classes.

likelihood, without affecting the ground-truth classification. For a source sample $(\mathbf{x}_s, y_s) \in \mathcal{D}_s$, we minimize the negative log-likelihood *w.r.t.* unknown logit to increase the unknown probability,

$$\mathcal{L}_{\text{UAT}}(f(\mathbf{x}), y) = \mathcal{L}_{\text{CE}}(f(\mathbf{x}), y) - \log \frac{\exp(f_u(\mathbf{x}))}{\sum_{c \in |\mathcal{C}_s|+1, c \neq y} \exp(f_c(\mathbf{x}))}, \quad (2)$$

where $f_u(\mathbf{x})$ is the unknown's logit. Eq. 2 makes the unknown class probability respond to any input sample, irrespective of its class label. Since the learning process is driven by the cross-entropy loss related to the ground-truth category, Eq. 2 does not hurt the performance of known classes.

## 3 Proposed Method

We propose CODA (Figure 2), a simple two-stage OTDG framework for discovering unknown classes with the help of known ones. The specific implementation of the two stages is described as follows.

### 3.1 Training-Time Source Compaction

The source compaction stage preserves sufficient space for the upcoming unknown classes without using real target data during training. Since we have no a *priori* knowledge or assumptions about the characteristics (*e.g.,* number and attribute) of unknown classes, it is prohibitively difficult to learn meaningful representations beforehand. To begin with, we make a mild assumption that the unknown classes should be far away from all known classes in the embedding space (category shift). In practice, there are two choices for accommodating these unknown classes: (1) low-density region (Figure 1 (b)), and (2) between-class region (Figure 1 (c)). The former is intuitive and could be achieved via Eq. 2, which naturally corresponds to these regions. However, the multi-modal structure of unknown-class data is underspecified, *i.e.,* different unknown classes should not occupy the same region. By contrast, the latter is able to implicitly separate different unknowns and model the relationships between known and unknown classes. To make the model expandable, our key idea

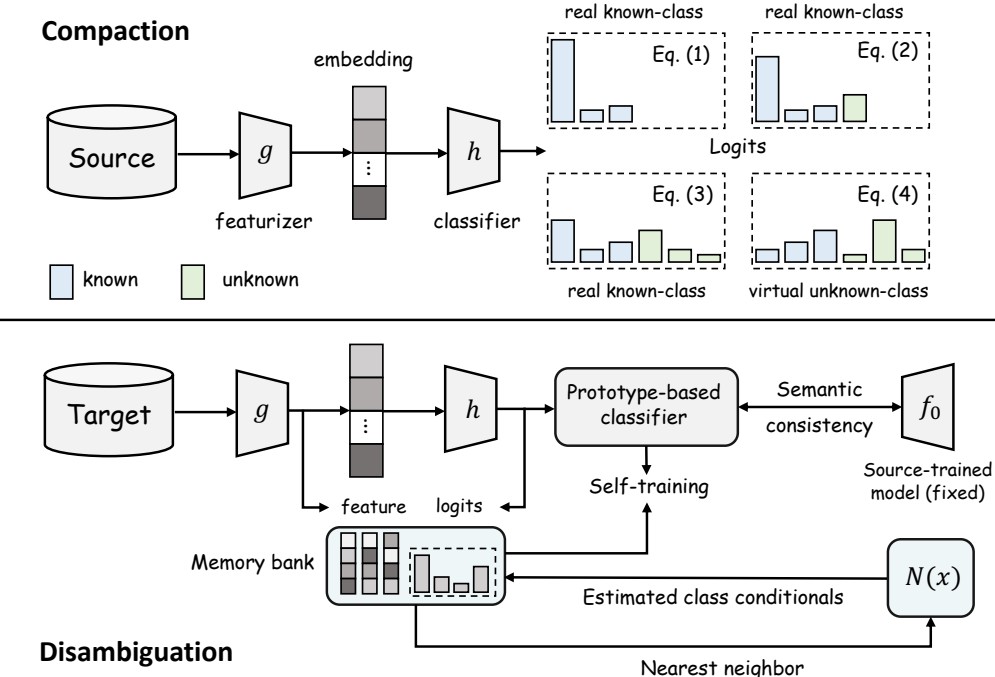

Figure 2: Overview of the proposed CODA, which consists of two novel components: (1) Training-time source **compaction** to make the model expandable for open classes; (2) Test-time target **disambiguation** to discriminate the decision boundaries with a test-time training objective.

is to compact the source embedding space by *(i)* making real known samples give response to both known and unknown classes, and *(ii)* inserting virtual unknown samples between known-class pairs.

**Optimization on real known-class samples.** In addition to the original source classes, we introduce a set of virtual classes $\mathcal{C}_v$ within the source embedding space to mimic the existence of unknown classes. To accommodate the virtual unknown classes, we regularize the embedding space by pushing known-class samples closer to the decision boundaries:

$$\mathcal{L}_{\text{real}}(f(\mathbf{x}), y) = \mathcal{L}_{\text{CE}}(f(\mathbf{x}), y) - \log \frac{\exp(f_{\hat{u}}(\mathbf{x}))}{\sum_{c \in |\mathcal{C}_s| + |\mathcal{C}_v|, c \neq y} \exp(f_c(\mathbf{x}))}, \quad (3)$$

where $f_{\hat{u}}(\mathbf{x})$ is the unknown's logit, having the largest activation among $\mathcal{C}_v$.

**Optimization on virtual unknown-class samples.** Despite the activation of the network's response to the introduced unknown classes, the "winner-takes-all" nature of softmax-based classification still leaves the unknown category unable to compete effectively with known ones (cf. Figure 2). To mimic test environments, we synthesize virtual unknown samples. Technically, we synthesize unknown samples at the *feature* level for model regularization, without using external data, *i.e.,* dashed points in Fig. 1 (c). We introduce manifold mixup [64] for synthesizing unknowns in the between-class regions, which are less confident for current decision boundaries. For two random samples $\mathbf{x}_i$ and $\mathbf{x}_j$ from different classes, we mix their embedding features as: $\hat{\mathbf{z}} = \mu \cdot g(\mathbf{x}_i) + (1 - \mu) \cdot g(\mathbf{x}_j)$, where $\mu$ is the mixing coefficient. The optimization objective for the synthesized unknown $\hat{\mathbf{z}}$ is defined as:

$$\mathcal{L}_{\text{virtual}}(h(\hat{\mathbf{z}}), \hat{y}) = \mathcal{L}_{\text{CE}}(h(\hat{\mathbf{z}}), \hat{y}) - \log \frac{\exp(h_{k'}(\hat{\mathbf{z}}))}{\sum_{c \in |\mathcal{C}_s| + |\mathcal{C}_v|, c \neq \hat{y}} \exp(h_c(\hat{\mathbf{z}}))}, \quad (4)$$

where $\hat{y}$ represents the label of $\hat{z}$ regarding unknown class, *i.e.,* having the largest activation among $\mathcal{C}_v$. $h_{k'}(\hat{\mathbf{z}})$ is the known's logit, having the largest activation among $\mathcal{C}_s$. The first term in Eq. 4 is a standard self-training loss. Similar to Eq. 3, the second term activates the response of $\hat{\mathbf{z}}$ to its most related known class. In essence, apart from the standard classification loss (the first term in Eq. 3 and Eq. 4), we activate the response of the real known class towards unknowns (Eq. 3) and the response of the virtual unknown class towards the known ones (Eq. 4).

## 3.2 Test-Time Target Disambiguation

Although we have allocated the embedding space for unknown classes, how to deploy the source-trained model on real test data is yet to be thoroughly studied. In particular, we identify two major challenges: (i) *optimality gap* between source and target domains, and (ii) *catastrophic forgetting* in open and dynamic test environments. In OTDG, we define the optimality gap as follows.

**Definition 1 (Optimality Gap)** *Let $\mathcal{H} \subseteq \{h | h : \mathcal{Z} \mapsto \mathcal{Y}\}$ be the hypothesis class. $\varepsilon_S(\cdot)$ and $\varepsilon_T(\cdot)$ denote the expected risk on source and target domains. For any hypothesis $h$, we have $\varepsilon_T(h^t) < \varepsilon_T(h^*)$, where $h^* = \arg\min_{h \in \mathcal{H}} \varepsilon_S(h) + \varepsilon_T(h)$ and $h^t = \arg\min_{h \in \mathcal{H}} \varepsilon_T(h)$.*

Definition 1 suggests that it is not feasible to find a *universal* optimal classifier that applies to both source and target domains. In that sense, the classifier, initially trained on source data, needs additional refinement to adapt effectively to the target patterns. Technically, we resort to TTA [34] to mitigate the above issues using unlabeled test data in an online manner. Conventional training-based TTA methods [67] usually need batches of data for self-training (*e.g.* entropy minimization) and/or heuristic self-supervision tasks [10]. On the other hand, training-free methods [23] require expensive tweaking of the threshold and only bring a marginal performance gain. Moreover, these approaches struggle to handle open-class scenarios, making them susceptible to negative transfer (*i.e.* semantic misalignment). To address these challenges, our proposed target disambiguation stage aligns the unlabeled target samples to their corresponding class patterns through the following process.

We construct a memory bank $\mathbb{S} = \{\mathbb{S}^1, \cdots, \mathbb{S}^{|\mathcal{C}_s| + |\mathcal{C}_v|}\}$ for memorizing the embedding $\mathbf{z}$ and logits $f(\mathbf{x})$ (or $h(\mathbf{z})$) of target samples. We compute a set of class prototypes $\{\mathbf{p}_k\}_{k=1}^{|\mathcal{C}_s| + |\mathcal{C}_v|}$ based on logits in $\mathbb{S}$. Following [23, 24], the memory bank is initialized with the weights of the linear classifier. Let $\mathcal{N}(\mathbf{x})$ be the Nearest Neighbor (NN) of $\mathbf{x}$ in $\mathbb{S}$. For each test sample $\mathbf{x}$[3], we search its NN in $\mathbb{S}$ by: $\mathcal{N}(\mathbf{x}) := \{\mathbf{z} \in \mathbb{S} | \text{sim}(g(\mathbf{x}), \mathbf{z}) \leq \theta_{\text{NN}}\}$, where $\text{sim}(\cdot)$ is the cosine similarity and $\theta_{\text{NN}}$ is a threshold to control the number of NN. The model predictions would be given by the similarity between sample embedding and the class prototype, *i.e.,* $p(y|\mathbf{x}) \propto \text{sim}\langle \mathbf{p}_k, \mathbf{z} \rangle$. Formally, for $\mathbf{z} \in \mathcal{N}(\mathbf{x})$, the likelihood of the prototype-based classifier assigning $\mathbf{z}$ to the $k$-th class can be calculated as follows:

$$p(y = k | \mathbf{z}) = \frac{\exp(-\text{sim}(h(\mathbf{z}), \mathbf{p}_k)/\tau)}{\sum_c \exp(-\text{sim}(h(\mathbf{z}), \mathbf{p}_c)/\tau)}, \ k = 1, 2, ..., |\mathcal{C}_s| + |\mathcal{C}_v|, \tag{5}$$

where $\tau$ is a temperature scaling parameter. We estimate the class conditionals with $\mathcal{N}(\mathbf{x})$ as:

$$\hat{\mathbf{p}}_k = \frac{1}{|\mathcal{N}(\mathbf{x})|} \sum_{\mathbf{z} \in \mathcal{N}(\mathbf{x})} \mathbb{1}(\arg\max_c p(c|\mathbf{z}) = k), \tag{6}$$

where $\mathbb{1}(\cdot)$ is an indicator function. Then, we can update the global class prototype computed from the whole $\mathbb{S}$ in a moving-average style,

$$\mathbf{p}_k \leftarrow \mu \mathbf{p}_k + (1 - \mu)\hat{\mathbf{p}}_k, \tag{7}$$

where $\mu \in [0, 1]$ is a preset scalar and $\hat{\mathbf{p}}_k$ can be regarded as the local class prototype. Given a batch of test samples $\mathcal{B}_t$, we use self-training for model updating ($g$ and $h$), *i.e.,* minimizing the cross-entropy loss between classifier's prediction $f(\mathbf{x})$ and the estimated class prior distribution $\mathbf{p}_k$:

$$\mathcal{L}_{\text{ST}}(\mathbf{x}) = \frac{1}{|\mathcal{B}_t|} \sum_{x \in \mathcal{B}_t} \mathcal{L}_{\text{CE}}(\mathbf{p}_k, f(\mathbf{x})), \tag{8}$$

**Semantic consistency.** To resist catastrophic forgetting during the online adaptation process, we enforce semantic consistency between the output of $f_0$ (source-trained model) and $f_I$ (target model) by optimizing the cross-entropy loss between their predictions:

$$\mathcal{L}_{\text{SC}}(\mathbf{x}) = -\sigma(f_0(\mathbf{x})) \log \sigma(f_I(\mathbf{x})), \tag{9}$$

where $I$ represents the number of iterations and $f_0$ is fixed throughout the testing phase.

**Reliable sample selection.** In the early stage of training, the estimation of target pseudo labels may be unreliable and thus is risky to error accumulation. To improve the quality of pseudo labels and

---

[3]We omit the subscript $t$ for simplicity.

reduce the influence of false class estimations, we introduce an entropy-based weighting strategy to select reliable samples. Specifically, we define the scoring function as follows:

$$S(\mathbf{x}) = \mathbb{1}(H(\mathbf{x}) < \theta_0) \exp(\theta_0 - H(\mathbf{x})), \tag{10}$$

where $H(\cdot)$ is the Shannon entropy of sample and $\theta_0$ is a pre-defined threshold. In this way, we can allocate larger weights to target samples with lower uncertainties and smaller weights to those with higher uncertainties, effectively prioritizing more confident predictions.

**Test-time training objective.** Formally, the overall optimization objective can be formulated as:

$$\mathcal{L}_{\text{TTT}}(\mathbf{x}) = S(\mathbf{x})\mathcal{L}_{\text{ST}}(\mathbf{x}) + \lambda \mathcal{L}_{\text{SC}}(\mathbf{x}), \tag{11}$$

where $\lambda$ is a trade-off parameter. The proposed $\mathcal{L}_{\text{TTT}}$ embraces the complementary strengths of parametric (softmax-based) and non-parametric (prototype-based) classifiers.

## 4 Experiments

### 4.1 Setup

**Benchmarks.** We conduct extensive experiments on four standard DG benchmarks to verify the effectiveness of CODA. **(1) PACS** [28] has 9,991 images and presents remarkable distinctions in image styles. It is comprised of four domains each with seven classes, *i.e., Photo*, *Art Painting*, *Cartoon*, and *Sketch*. Dog, elephant, giraffe, and guitar are used as $\mathcal{C}_s$ while the remaining 3 classes are $\mathcal{C}_t^u$. **(2) Office-Home** [63] is gathered from both office and home environments, and its domain shifts originate from variations in viewpoint and image style. It has 15,500 images of 65 classes from four domains, *i.e., Artistic*, *Clipart*, *Product*, and *Real World*. Arranged in alphabetical order, the initial 15 classes are designated as $\mathcal{C}_s$, and the remaining 50 classes are assigned to $\mathcal{C}_t^u$. **(3) Office-31** [48] encompasses 31 classes harvested from three distinct domains: *Amazon*, *DSLR*, and *Webcam*. The 10 classes shared by Office-31 and Caltech-256 [18] are used as $\mathcal{C}_s$. In alphabetical order, the final 11 classes, combined with $\mathcal{C}_s$, constitute $\mathcal{C}_t$. **(4) Digits**, a dataset varying in background, style, and color, encompasses four domains of handwritten digits, including *MNIST*[26], *MNIST-M*[17], *SVHN*[42], *USPS*[22], and *SYN* [17]. We utilize *MNIST* as the source domain, while the other datasets serve as target domains. Numbers from 0 to 4 make up $\mathcal{C}_s$.

**Evaluation Protocols.** In line with prior works [5, 90, 77], we utilize the H-score ($hs$) [16] as our main evaluation criterion. The $hs$ score balances the significance between known and unknown classes, emphasizing that both groups should have high and equivalent accuracy. Additionally, the $hs$ score circumvents the trivial solution where a model would predict all samples as known classes. The classification accuracy for both known ($acc_k$) and unknown ($acc_u$) classes are also reported.

**Implementation Details.** For PACS, Office-Home, and Office-31, we employ ResNet-18 [19], pre-trained on ImageNet, as the backbone network. For Digits, we employ the LeNet [25] with the architecture arranged as *conv-pool-conv-pool-fc-fc-softmax*. The training is performed using SGD with a momentum of 0.9 for 100 epochs, and we set the batch size to 64. Our experiments are built upon Dassl [87] (a PyTorch toolbox developed for DG), covering aspects of data preparation, model training, and model selection. We report the means over 5 runs with different random seeds.

### 4.2 Baselines

In experiments, we empirically compare CODA against five types of baselines: **(1) OSDG:** CM [90] and One Ring-S [77]. **(2) OSDA:** OSBP [49] and ROS [5]. **(3) OD:** MSP [20], LogitNorm [71], and DICE [56]. **(4) SFDA:** SHOT [35] and AaD [76]. **(5) TTA:** TTT [58], Tent [67], MEMO [80], TAST [24], and FAU [69]. Since TTA methods are incapable of directly handling unknown-class samples, we adopt the approach from [90], using the entropy of the softmax output as the final prediction. For ERM [60], we follow the same strategy for identifying unknowns.

### 4.3 Results

Our results are summarized in Table 2. For each dataset, CODA outperforms all compared methods by a considerable margin in terms of $hs$. For instance, in comparison to the previous best-performing OSDG baseline [77], CODA achieves increases in $hs$ by 16.8% for PACS, 4.0% for Office-Home,

Table 2: Accuracy (%) on PACS, Office-Home, Office-31, and Digits datasets.

| Regime | Method | PACS | | | Office-Home | | | Office-31 | | | Digits | | |
|---|---|---|---|---|---|---|---|---|---|---|---|---|---|
| | | $acc_k$ | $acc_u$ | $hs$ | $acc_k$ | $acc_u$ | $hs$ | $acc_k$ | $acc_u$ | $hs$ | $acc_k$ | $acc_u$ | $hs$ |
| OSDA | OSBP [49] | 40.6 | 49.5 | 44.6 | 47.1 | 66.9 | 55.3 | 75.8 | 84.3 | 77.7 | 35.6 | 70.6 | 40.5 |
| | ROS [5] | 35.6 | 66.4 | 46.4 | 50.8 | 77.5 | 60.8 | 71.7 | 80.0 | 75.6 | 20.1 | 48.6 | 34.9 |
| OD | MSP [20] | 38.9 | 62.5 | 46.4 | 52.7 | 75.6 | 62.0 | 49.7 | 89.2 | 63.8 | 17.2 | 87.1 | 28.8 |
| | LogitNorm [71] | 35.1 | 47.6 | 38.3 | 56.3 | 56.5 | 56.1 | 41.0 | 71.2 | 52.1 | 26.8 | 51.2 | 35.2 |
| | DICE [56] | 44.0 | 53.4 | 49.2 | 61.5 | 58.8 | 59.9 | 72.8 | 61.1 | 66.4 | 35.0 | 47.6 | 40.3 |
| SFDA | SHOT [35] | 51.2 | 34.9 | 40.8 | 52.5 | 32.4 | 44.3 | 84.8 | 60.2 | 70.4 | 27.4 | 20.3 | 23.3 |
| | AaD [76] | 45.1 | 40.0 | 42.0 | 59.4 | 58.7 | 58.9 | 70.1 | 85.3 | 76.9 | 25.6 | 26.9 | 26.2 |
| TTA | TTT [58] | 36.9 | 44.6 | 38.9 | 52.0 | 45.9 | 47.2 | 35.4 | 79.6 | 49.0 | 40.1 | 41.1 | 40.6 |
| | Tent [67] | 25.2 | 43.1 | 31.7 | 33.6 | 45.9 | 38.7 | 56.0 | 85.1 | 67.5 | 27.2 | 41.1 | 32.7 |
| | MEMO [80] | 37.9 | 52.3 | 44.5 | 49.0 | 55.6 | 52.1 | 59.8 | 72.7 | 65.6 | 21.7 | 56.1 | 31.3 |
| OSDG | ADA [66] | 54.2 | 30.9 | 36.4 | 67.9 | 25.4 | 36.2 | 85.6 | 25.2 | 38.7 | 57.2 | 15.1 | 20.1 |
| | ADA+CM [90] | 56.4 | 45.6 | 43.0 | 65.0 | 40.4 | 48.5 | 83.0 | 34.5 | 48.5 | 49.2 | 52.1 | 39.9 |
| | MEADA [81] | 54.1 | 31.4 | 36.2 | 67.6 | 25.7 | 36.4 | 85.8 | 25.1 | 38.6 | 57.6 | 29.8 | 30.4 |
| | MEADA+CM [90] | 54.3 | 46.6 | 42.7 | 64.9 | 40.5 | 49.6 | 82.8 | 41.1 | 54.7 | 52.3 | 46.1 | 38.7 |
| | One Ring-S [77] | 43.7 | 49.4 | 41.5 | 56.9 | 69.0 | 62.3 | 67.3 | 77.0 | 71.3 | 33.2 | 51.3 | 40.3 |
| OTDG | ERM [60] | 52.3 | 27.0 | 36.1 | 66.9 | 23.7 | 34.3 | 85.1 | 27.0 | 40.7 | 56.4 | 13.0 | 18.0 |
| | CODA (ours) | 54.3 | 63.8 | **58.3** | 59.7 | 74.6 | **66.3** | 87.5 | 75.4 | **81.0** | 31.5 | 60.1 | **41.3** |

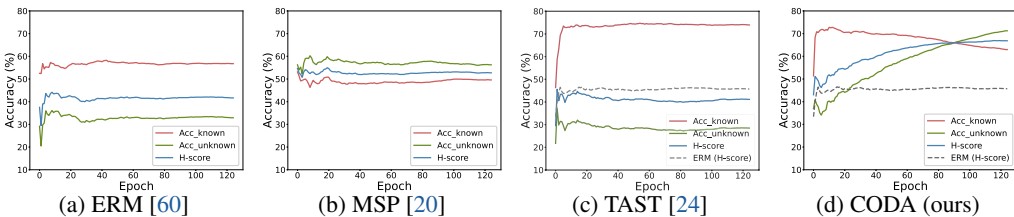

(a) ERM [60]  (b) MSP [20]  (c) TAST [24]  (d) CODA (ours)

Figure 3: Performance comparisons of different methods as testing proceeds on the PACS dataset.

9.7% for Office-31, and 1.0% for Digits. If we focus on the hard generalization tasks, such as PACS, CODA exhibits larger performance gains than on other tasks. Moreover, three trends can be observed: (1) Compared to OSDA and SFDA methods that usually optimize with target data offline, CODA achieves superior performance in an online adaptation manner. (2) The $acc_k$ and $acc_u$ of Tent [67] and LogitNorm [71] exhibit significant imbalance as both methods tend to predict all data as known-class samples (*i.e.* shortcut learning). This verifies the benefits of CODA in mitigating shortcut learning. (3) OD methods achieve very competitive results compared to other types of baselines. The rationale is that they usually do not involve test-time adjustment and thus have better stability. (4) The performance of different types of baseline methods varies across benchmarks. For instance, MEMO (TTA) achieves the second-best result in PACS but has relatively inferior performance in Office-Home and Digits. Instead, CODA exhibits consistent improvements on all benchmarks.

In addition, we provide performance comparisons of different methods (*i.e.,* ERM [60], MSP [20], TAST [24], and our CODA) as testing proceeds on the PACS dataset (trained on domain *Art Painting*). To facilitate a fair comparison, both MSP and TAST will apply to the models that have been trained by our source compaction stage. Figure 3 shows that ERM and TAST substantially increase $acc_k$ and maintain it at a very high level, which severely impedes the improvements of $hs$. Interestingly, as the number of testing epochs increases, TAST underperforms

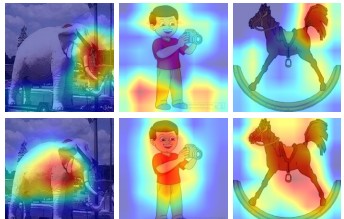

Figure 4: *Top:* ERM. *Bottom:* CODA.

compared to ERM. By contrast, CODA dynamically harmonizes the relations between $acc_k$ and $acc_u$ (*i.e.,* suppresses $acc_k$ and hence allows $acc_u$ to grow), which is reflected by the monotonic increase of $hs$. Figure 4 shows Grad-CAM [52] visualizations of baseline (ERM) and our method (CODA) on the PACS dataset. We can see that the hot zones activated by CODA are more complete and reasonable, providing a reliable semantic understanding of the foreground object.

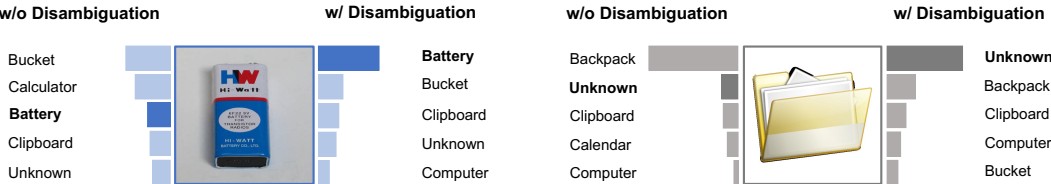

Figure 5: Predictions from models trained with and without target disambiguation.

## 4.4 Ablation Studies

**Ablations of key components in CODA.** We carry out ablation studies in Table 3, evaluating the effect of source compaction (SC) and target disambiguation (TD) proposed in CODA. When we exclude SC from CODA, model predictions are made based on the entropy of the output in conjunction with a predetermined threshold [90]. From the table, we can observe that adding SC and TD could improve the generalization per-

Table 3: Ablation of CODA on four classification benchmarks. $hs$ (%) is reported.

| SC | TD | PACS | Office-Home | Office-31 | Digits |
|----|----|------|-------------|-----------|--------|
| ×  | ×  | 36.1 | 34.3 | 40.7 | 18.0 |
| ✓  | ×  | 51.2 | 64.8 | 75.3 | 39.0 |
| ×  | ✓  | 49.8 | 61.7 | 72.3 | 37.5 |
| ✓  | ✓  | 58.3 | 66.3 | 81.0 | 41.3 |

formance, which verifies their individual contributions for solving domain shifts and open classes. Moreover, our method that integrates both SC and TD achieves the best performance, revealing the synergistic effect between the two components.

**Analysis on target disambiguation.** In Figure 5, we plot the predictions on known- and unknown-class test samples from the models trained with and without target disambiguation. The left image is a `battery` (known), and the right image is a `clipboard` (unknown). For the known classes, a model lacking disambiguation often makes uncertain predictions, especially for hard samples that bear high resemblance to other classes. For the unknown class, a model without disambiguation tends to give a high response to an incorrect class and suppress responses to other classes. In contrast, our target disambiguation stage can achieve more accurate predictions by recovering the semantic relationships among classes from unlabeled data, thereby enhancing the model's generalization performance under both domain shift and open classes. In Figure 6 (a)-(b), we investigate the the impact of varying the test batch size on three methods, TAST [24], FAU [69], and CODA (ours). As the batch size varies, CODA consistently delivers superior performance compared to TAST and FAU, revealing the advantages of the proposed online adaptation strategy.

**Analysis on unknown-aware training objective.** We empirically compare different unknown-aware training objectives discussed in Section 2 and Section 3.1, *i.e.,* Eq.(1)-(4) and their combinations. For a fair comparison, we do not involve any TTA strategies including our target disambiguation. The results are reported in Table 4. We can observe

Table 4: Analysis on unknown-aware training objective.

| Method | PACS | Office-Home | Office-31 | Digits |
|--------|------|-------------|-----------|--------|
| Eq. 1 | 36.1 | 34.3 | 40.7 | 18.0 |
| Eq. 2 | 41.1 | 58.9 | 63.0 | 39.3 |
| Eq. 3 | 46.8 | 62.4 | 72.6 | 38.6 |
| Eq. 4 | 43.2 | 60.3 | 70.7 | 37.4 |
| Eq. 3 + Eq. 4 | 51.2 | 64.8 | 75.3 | 39.0 |

that our full source compaction is clearly better than its variants, revealing the superiority of our optimization procedures on both real known-class samples and virtual unknown-class samples.

**Analysis on unknown classes.** In Figure 6 (c)-(d), we study the impact of varying the number of known classes on three methods, ERM [60], One Ring-S [77], and CODA (ours). Note that the total number of classes (*i.e.* $|\mathcal{C}_s \cup \mathcal{C}_t|$) remains unchanged. Even when the number of known classes is small, CODA still exhibits superior performance. This advantage remains consistent as the number of known classes changes. Consequently, CODA is capable of handling extreme scenarios.

**Feature visualization.** We use $t$-SNE [59] to visualize the features of four models on Office-31 dataset, *i.e.,* ERM, One Ring-S, Source Compaction, and CODA (full). The results are depicted in Figure 7, where various colors, excluding gray, signify different known classes, and gray points represent all unknown classes. It is noteworthy that the embedding features learned by two baselines (ERM and One Ring-S) fail to present a clear separation, resulting in ambiguous boundaries among

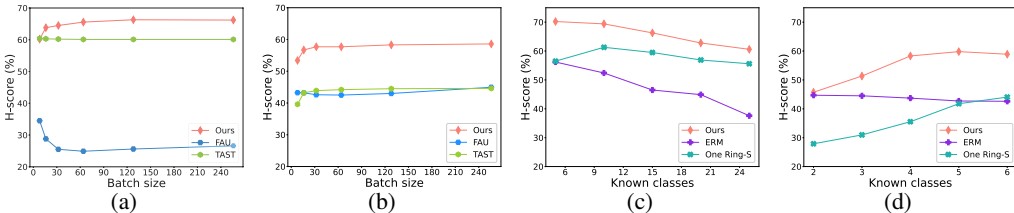

Figure 6: (a)-(b) The influence of test batch size. (c)-(d) The influence of varying the number of known classes. Figures (a) and (c) are plotted based on the Office-Home dataset, while figures (b) and (d) are derived from the PACS dataset.

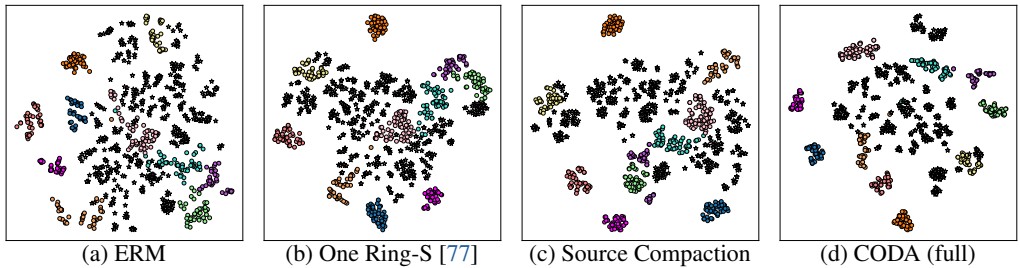

    (a) ERM          (b) One Ring-S [77]      (c) Source Compaction      (d) CODA (full)

Figure 7: t-SNE visualization of the learned features on the Office-31 dataset.

different classes particularly between known and unknown ones. Instead, CODA can learn the intrinsic structure ("manifold") of the target samples, providing more discriminable clustering patterns.

## 5 Related Work

**Domain Generalization (DG).** DG is concerned with the training of a model using data from either multiple or a single source domain, with the goal of achieving good generalization to previously unseen target domains. Mainstream approaches usually involve invariant learning and robust learning with elaborate training objectives. Based on the focus of this paper, we classify existing methods into three categories and provide descriptions as follows. **(1) Closed-set DG.** Existing methods can be roughly divided into four categories: feature matching-based [32, 31, 40, 91, 83], decomposition-based [46, 45, 13, 39, 54, 36, 78], augmentation-based [65, 85, 86, 74, 41, 88, 8], and meta-learning-based approaches [29, 33, 30, 37, 7]. **(2) Open-set DG.** It is worth noting that very few works have delved into the exploration of DG in open-set scenarios. A handful of recent studies [53, 90, 77, 9] started to consider the existence of both known and unknown classes in novel DG settings, such as Open-Set DG (OSDG) [90, 77]. For example, Yang *et al.* [77] hold the view that any category other than the ground-truth can be considered as part of the unknown classes. Chen *et al.* [9] first activate the unknown's logit via an unknown-aware training loss and then introduce a test-time adjustment strategy to refine the model prediction. Zhu *et al.* [90] rely on heuristic thresholding mechanisms to distinguish known- and unknown-class samples. **(3) DG by test-time adaptation.** According to Dubey *et al.* [14], a model trained solely on source data will inevitably experience an "adaptivity gap" when it is directly employed on target domains, emphasizing the necessity of on-target adaptation. Grounded on this insight, several recent works [79, 73, 72, 11] resort to TTA for mitigating the adaptivity gap, such as adaptive risk minimization [79], energy-based sample adaptation [72], and improved self-supervised learning tasks [11].

**Test-Time Adaptation (TTA).** TTA [34] is an emerging paradigm that has demonstrated immense potential in adapting pre-trained models to unlabeled data during the testing phase. A plethora of approaches [35, 58, 67, 23, 43, 10, 80, 69] have been developed to improve the predictions of source-trained models on target domain with online training/adaptation strategies. TTT [58] introduces self-supervised learning tasks (*e.g.* rotation classification) to both source and target domains. Tent [67] leverages the batch statistics of the target domain and optimizes the channel-wise affine parameters by entropy minimization. T3A [23] proposes to use class prototypes for adjusting predictions and introduces a support set to memorize representative and reliable samples. TAST [24] improves T3A by proposing a nearest neighbor information induced self-training framework.

**Out-of-Distribution Detection (OD).** OD [75], which seeks to identify novel instances that the model has not encountered during training, is close to OTDG setting. Prevailing OD methods center on creating OOD scoring functions, for example, confidence-based techniques [4, 20, 21], distance-based scores [27, 51, 57], and energy-based scores [38, 55]. Although promising, the OD approach is limited to binary classification problems and lacks the capability to effectively explore domain shift and open class challenges in the test data.

There are also other topics related to open-world machine learning [44] that bear certain relevance to our work, including open-set recognition [50, 62], novel class discovery [15, 61], zero-shot learning [47, 70], and class-incremental learning [6, 82], to name a few. Compared to previous methods, our work addresses two types of open-world situations (*i.e.,* domain shift and open class), supporting generalization capabilities consistently throughout the training and inference phases.

## 6   Conclusion

In this paper, we solve the problem of OTDG where both domain shift and open classes may concurrently arise on the unseen test data. We introduce a two-stage framework (CODA) for efficiently learning what we don't know in the wild. At the training stage, we compact the embedding of source known classes and thus reserve space for target unknown classes. In the testing phase, we introduce a training objective to mitigate the optimality gap between domains while avoiding catastrophic forgetting. Empirically, CODA achieves superior performance on standard DG benchmarks.

## Acknowledgement

This work was partially supported by Hong Kong Research Grants Council under Collaborative Research Fund (Project No. HKU C7004-22G).

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
