# CODA: Generalizing to Open and Unseen Domains with Compaction and Disambiguation
## *(Supplementary Materials)*

**Chaoqi Chen**[1][*] **Luyao Tang**[2][*] **Yue Huang**[2] **Xiaoguang Han**[3][†] **Yizhou Yu**[1][†]

[1]The University of Hong Kong    [2]Xiamen University
[3]The Chinese University of Hong Kong (Shenzhen)
cqchen1994@gmail.com, lytang@stu.xmu.edu.cn, yhuang2010@xmu.edu.cn
hanxiaoguang@cuhk.edu.cn, yizhouy@acm.org

This document provides more details of our approach and additional experimental results.

## A Implementation Details

### A.1 Pipeline

We summarize the pipeline of the proposed CODA in Algorithm 1, which consists of two stages, *i.e.,* source compaction and target disambiguation.

### A.2 Model selection

Following `Dassl`[3] and `OneRing`[4], we trained the model for 100 epochs and used the last-step checkpoint for evaluation, which is fair and reasonable in the context of domain generalization. Since previous works do not provide a dedicated train-validation split, it is not possible to create a separate validation set from the training dataset for model selection.

### A.3 The choice of hyperparameters

**Source Compaction.** During the training phase, we set the number of virtual classes to be equal to the number of source classes, $|\mathcal{C}_v| = |\mathcal{C}_s|$, and the balancing parameter $\gamma$, which controls the trade-off between $\mathcal{L}_{\text{real}}$ and $\mathcal{L}_{\text{virtual}}$, is set to 0.0005. For the mixing coefficient $\mu$, we set it to 0.5.

**Target Disambiguation.** For $\mathcal{N}(\mathbf{x})$, we set the number of nearest neighbors to 10. For threshold $\theta_0$ in $S(\mathbf{x})$, we set it to $0.5 \ln (|\mathcal{C}_s| + |\mathcal{C}_v|)$. For $\mathcal{L}_{\text{TTT}}$, we set the balancing parameter $\lambda$ to 0.1.

Other hyperparameters have been clarified in the main paper.

## B Experimental Results

### B.1 Ablation studies

In the main paper, we have extensively compared the proposed CODA with different types of baseline approaches: Open-Set Domain Adaptation (OSDA) [5, 1], OD [2, 10, 7], Source-Free Domain Adaptation (SFDA) [4, 11], Test-Time Aadaptation (TTA) [8, 9, 13], and Open-Set Domain

---

[*]First two authors contributed equally.

[†]Corresponding authors.

[3]https://github.com/KaiyangZhou/Dassl.pytorch

[4]https://github.com/Albert0147/OneRing_SF-OPDA

**Algorithm 1** The pipeline of proposed CODA.

---

**Input:** Source domain $\mathcal{D}_s = \{(\mathbf{x}_s^i, y_s^i)\}_{i=1}^{n_s}$, featurizer $g : \mathcal{X} \mapsto \mathcal{Z}$, and predictor $h : \mathcal{Z} \mapsto \mathcal{Y}$.
**Output:** The label predictions for unseen target domains $\mathcal{D}_t = \{(\mathbf{x}_t^j)\}_{j=1}^{n_t}$.
**while** *train* **do**
   | Train $g$ and $h$ with $\mathcal{L}_{\text{real}} + \mathcal{L}_{\text{virtual}}$ using $\mathcal{D}_s$.
**end**

Initialize the memory bank $\mathbb{S} = \{\mathbb{S}^1, \cdots, \mathbb{S}^{|\mathcal{C}_s|+|\mathcal{C}_v|}\}$ and class prototypes $\{\mathbf{p}_k\}_{k=1}^{|\mathcal{C}_s|+|\mathcal{C}_v|}$.
**for** $\mathbf{x}_t^j \in \mathcal{D}_t$ **do**
   Obtain its embedding feature $\mathbf{z}_t^j = g(\mathbf{x}_t^j)$.
   Compute its nearest neighbor $\mathcal{N}(\mathbf{x}_t^j)$ in $\mathbb{S}$, based on $\mathcal{N}(\mathbf{x}) = \{\mathbf{z} \in \mathbb{S}|\text{sim}(g(\mathbf{x}), \mathbf{z}) \leq \theta_{\text{NN}}\}$.
   **for** $\mathbf{z} \in \mathcal{N}(\mathbf{x}_t^j)$ **do**
     | Compute the likelihood of assigning $\mathbf{z}$ to the $k$-th class: $p(y = k|\mathbf{z}), k = 1, 2, ..., |\mathcal{C}_s| + |\mathcal{C}_v|$.
   **end**
   Estimate the class conditionals $\hat{\mathbf{p}}_k$ with $\mathcal{N}(\mathbf{x}_t^j)$, based on Eq. (6).
   Update the global class prototype computed from the whole $\mathbb{S}$: $\mathbf{p}_k \leftarrow \mu\mathbf{p}_k + (1 - \mu)\hat{\mathbf{p}}_k$.
   Compute the test-time training objective $\mathcal{L}_{\text{TTT}}(\mathbf{x}_t^j) = S(\mathbf{x}_t^j)\mathcal{L}_{\text{ST}}(\mathbf{x}_t^j) + \lambda\mathcal{L}_{\text{SC}}(\mathbf{x}_t^j)$.
**end**

---

Generalization (OSDG) [14, 12].According to Table 3 in the main paper, the results show that even when **Target Disambiguation (TD)** is not utilized, the performance of using only **Source Compaction (SC)** is still significantly better than all the compared methods. In this section, we extend our investigation by examining the impact of TD when combined with other state-of-the-art methods for open-set domain generalization (OSDG). Specifically, we consider the combination of One Ring-S [12], which is a leading OSDG method, with TD, TTA methods including TTT [8], Tent [9], and Memo [13], as well as SFDA methods such as SHOT [4] and AaD [11]. The results are shown in Table 1 and Table 2. The results clearly demonstrate that the benefits of TD surpass those of other TTA and SFDA methods. In particular, Tent [9] exhibits lower performance due to the imbalance between the accuracy of known classes ($acc_k$) and unknown classes ($acc_u$), indicating the challenges involved in applying TTA techniques in our specific task.

### B.2 Parameter sensitivity

We evaluate the sensitivity of CODA to hyper-parameters $N$ (the number of virtual classes) and $\gamma$ (balancing parameter that controls the trade-off between $\mathcal{L}_{\text{real}}$ and $\mathcal{L}_{\text{virtual}}$) on the PACS, Office-Home, and Office-31 datasets, respectively. The results are presented in Figure 1. It can be observed that the performance of CODA is not significantly affected by variations in different parameters. Specifically, the H-score remains relatively stable under a wide range of hyper-parameter values. These findings provide evidence for the strong efficacy and scalability of the proposed CODA method in diverse real-world scenarios.

### B.3 Visualization

Figure 2 shows Grad-CAM [6] visualizations of baseline (ERM) and our method (CODA) on the PACS dataset. Consistent with the observations reported in the main paper, the activated regions highlighted by CODA provide a comprehensive and accurate representation of the foreground object. These regions effectively capture the discriminative features and contribute to a more precise understanding of the object within the image.

## C  Limitation

While the proposed CODA approach treats all open classes as a single unknown class, it is important to acknowledge that in certain real-world scenarios, there may be a need to not only identify outliers but also uncover their inner semantic structures. In these cases, the current CODA framework may not be applicable as it focuses primarily on addressing the presence of unknown classes. Any modification

Table 1: Additional ablation studies. $hs$ (%) is reported.

| Method | PACS | Office-Home | Office-31 | Digits | Average |
|---|---|---|---|---|---|
| SC | 51.2 | 64.8 | 75.3 | 39.0 | 57.6 |
| + TTT [8] | 52.0 | 65.2 | 75.5 | 39.4 | 58.0 (+0.4) |
| + Tent [9] | 40.7 | 59.6 | 66.5 | 35.0 | 50.5 (-7.1) |
| + T3A [3] | 53.0 | 65.8 | 77.0 | 40.2 | 59.0 (+1.4) |
| + MEMO [13] | 53.9 | 65.0 | 77.4 | 40.5 | 59.2 (+1.6) |
| + SHOT [4] | 49.3 | 60.8 | 72.4 | 36.2 | 54.7 (-2.9) |
| + AaD [11] | 55.6 | 65.4 | 79.8 | 40.7 | 60.4 (+2.8) |
| + TD (ours) | 58.3 | 66.3 | 81.0 | 41.3 | **61.7** (**+4.1**) |

Table 2: Additional ablation studies. $hs$ (%) is reported.

| Method | PACS | Office-Home | Office-31 | Digits | Average |
|---|---|---|---|---|---|
| One Ring-S [12] | 41.5 | 62.3 | 71.3 | 40.3 | 53.9 |
| + TTT [8] | 42.2 | 61.8 | 71.6 | 40.2 | 54.0 (+0.1) |
| + Tent [9] | 28.3 | 30.0 | 68.8 | 34.5 | 40.4 (-13.5) |
| + T3A [3] | 47.4 | 65.0 | 72.1 | 40.8 | 56.3 (+2.4) |
| + MEMO [13] | 48.6 | 64.1 | 72.5 | 41.2 | 56.6 (+2.7) |
| + SHOT [4] | 39.5 | 60.9 | 66.8 | 37.4 | 51.2 (-2.7) |
| + AaD [11] | 47.3 | 63.2 | 75.0 | 41.4 | 56.7 (+2.8) |
| + TD (ours) | 55.5 | 63.2 | 76.9 | 40.0 | **58.9** (**+5.0**) |

or extension of the approach would be necessary to tackle the additional requirement of understanding the inner structures of outliers.

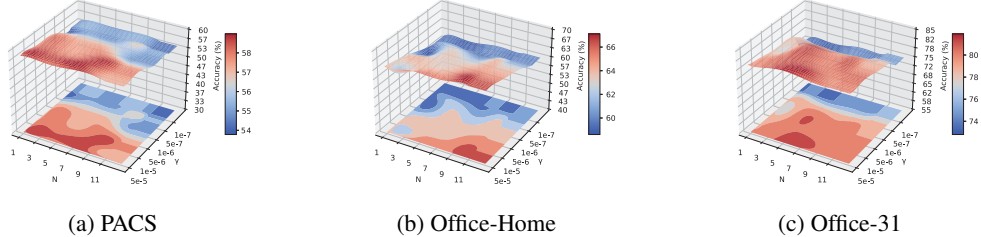

(a) PACS  (b) Office-Home  (c) Office-31

Figure 1: the sensitivity of CODA to hyper-parameters $N$ and $\gamma$.

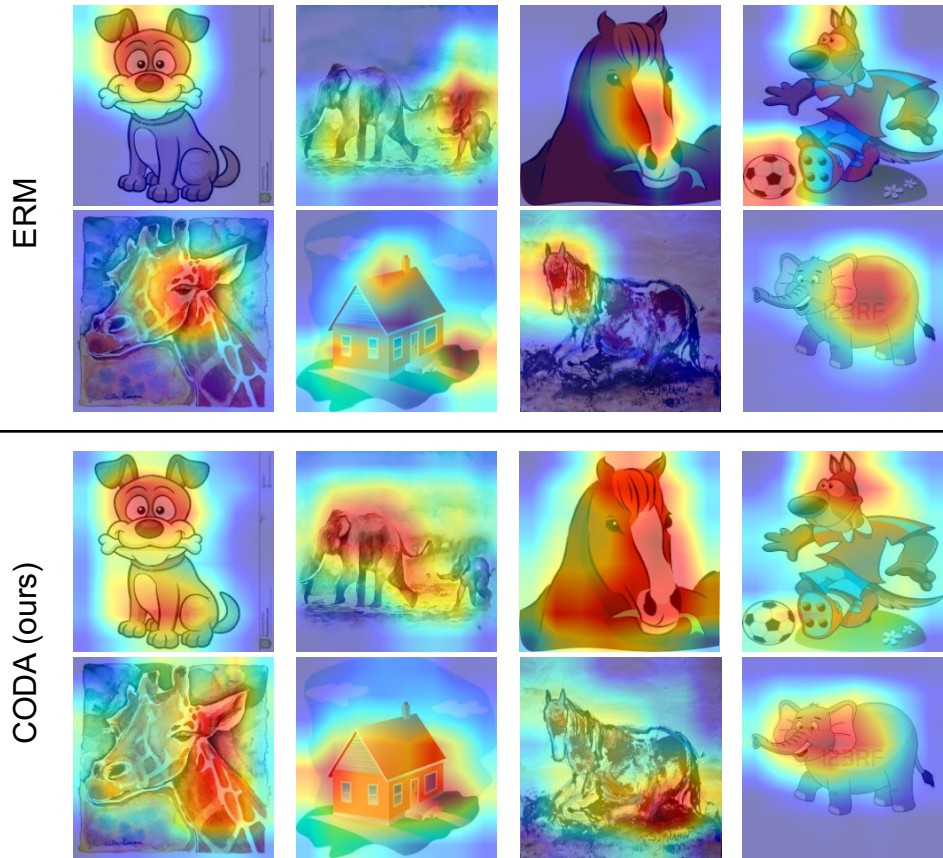

Figure 2: Gram-CAM visualizations of ERM (top) and CODA (bottom).