# OpenReview forum: "CODA: Generalizing to Open and Unseen Domains with Compaction and Disambiguation"
_NeurIPS.cc/2023/Conference — NeurIPS 2023 spotlight_

### Official Review · Reviewer_shJ8 · 2023-07-01

**Soundness:** 3 good
**Presentation:** 3 good
**Contribution:** 4 excellent
**Rating:** 8
**Confidence:** 4

**Summary:**

This paper proposes a two-stage framework for the task of open-class domain generalization. During training, it introduces virtual unknown classes for the learning objective to insert unknowns into the latent space by compacting the embedding space of known classes. During the test phase, it proposes a prototype-based test-time adaptation pipeline, which involves (1) consistency between the predicted class distributions and the estimated class conditionals; (2) class-wise sample reliability for ensuring the quality of target pseudo labels; (3) semantic consistency between the source-trained model and target model, to disambiguate the decision boundaries between known and unknown classes. Experimental results on different DG benchmarks demonstrate its effectiveness.

**Strengths:**

1. The paper studies a practical problem that has been rarely explored, open test-time domain generalization, which considers the co-occurrence of domain shift and open class. The comparison with other related settings is clearly stated and provides good intuitions.

2. The proposed framework, CODA, is novel and technically sound. The introduced compaction and disambiguation process is easy to implement, and the computational overhead is relatively small. Overall, the paper is well organized, and the advantage of CODA is clearly demonstrated.

3. The experiments confirm the effectiveness of the method. In particular, different types of baselines, such as DG, DA, TTA, and OOD detection, are explicitly compared. Ablations and visualization further justify the contributions of individual components.

**Weaknesses:**

1. There are some inconsistencies that need to be clarified. Firstly, the authors initially utilize the term 'adaptivity gap', but introduce a new terminology, 'optimality gap', in Definition 1. What are their distinction and correlation? Secondly, the proposed setting called OTDG appears to underscore the process of test-time operation. However, I found that the source training stage offers large improvements according to the reported results. As such, I question whether the use of 'test-time domain generalization' is fitting. Thirdly, in Figure 2, the authors use dashed points to signify virtual unknown samples. However, these samples are produced by manifold mixup at the feature level, which makes them distinct from the original data.

2. Although Figure 1 provides an intuitive comparison among different learning objectives, I struggle to fully comprehend why Equation (3) is superior to Equation (2). The advantages of employing multiple virtual prototypes versus a single virtual prototype warrant further discussion and comparison.

3. Basically, the target disambiguation is a prototype-based classification process. As the target distribution contains some open classes, how the disambiguation is done by only using the source-learned virtual prototypes? Moreover, could the proposed CODA be extended to a more challenging scenario that the open classes are further classified into different clusters?

4. How are the experiments conducted with different types of baselines, including aspects like train-validation split and model selection? Further clarification and details regarding the fairness of comparison should be provided.

**Questions:**

See weakness.

**Limitations:**

Limitations are discussed in the supplementary material.

---

> ### Author Rebuttal · Authors · 2023-08-08
>
> Thank you for your constructive feedback and for recognizing our novel contributions. We list our responses to your questions below.
>
> > **Q1. The authors initially utilize the term 'adaptivity gap', but introduce a new terminology, 'optimality gap', in Definition 1. What are their distinction and correlation?**
>
> The term 'adaptivity gap' was first introduced in [11]. In this work, we define a new terminology - optimality gap - to highlight the disparity between the target optimal classifier and the universal classifier. In essence, these two terminologies can be used interchangeably. For the sake of consistency, we will adopt the terminology from [11] in the final version.
>
> > **Q2. The proposed setting called OTDG appears to underscore the process of test-time operation. However, I found that the source training stage offers large improvements according to the reported results. As such, I question whether the use of 'test-time domain generalization' is fitting.**
>
> Thank you for pointing out this interesting problem. In this work, we focus on DG safety by solving two open-world issues in a unified framework: open class and adaptivity gap. The former is solved by unknown-aware training objectives, while the latter is addressed by test-time adaptation. These two components are complementary to each other and should not be regarded separately. We agree that the source training stage offers large improvements according to the reported results, but these empirical results do not affect the ultimate goal of OTDG.
>
> > **Q3. In Figure 2, the authors use dashed points to signify virtual unknown samples. However, these samples are produced by manifold mixup at the feature level, which makes them distinct from the original data.**
>
> We apologize for the confusion. You're correct; the dashed points represent virtual unknown samples created through manifold mixup. We will clarify this in the figure's caption and provide a more detailed description in the main text to avoid future misunderstandings.
>
> > **Q4. I struggle to fully comprehend why Equation (3) is superior to Equation (2). The advantages of employing multiple virtual prototypes versus a single virtual prototype warrant further discussion and comparison.**
>
> Thank you for raising this concern. Equation (3) leverages multiple virtual prototypes, which aim to provide a more comprehensive representation of the unknown-class manifold, capturing a wider range of potential variations. On the other hand, Equation (2) with a single virtual prototype may not sufficiently represent the diversity of unknown classes. In our experiments (see Table 4), we found that using multiple virtual prototypes often leads to more robust performance, especially in scenarios with a high diversity of unknown samples, such as PACS and Office-31. We will expand upon this reasoning and include additional comparisons in the revised manuscript to further elucidate the benefits of employing multiple virtual prototypes.
>
> > **Q5. How the disambiguation is done by only using the source-learned virtual prototypes?**
>
> The source-learned virtual prototypes serve as representative points in the feature space for unknown classes. In the case of OTDG, these prototypes help in identifying and differentiating between the known and unknown samples. Technically, during testing, the distance (i.e. Eq. 5) between a test sample's feature representation and these virtual prototypes can be computed. After that, we could discriminate the real known and unknown classes, i.e., disambiguation.
>
> > **Q6. Could the proposed CODA be extended to a more challenging scenario that the open classes are further classified into different clusters?**
>
> The CODA framework can indeed be extended to handle scenarios where open classes are further classified into distinct clusters. Leveraging our plug-and-play 'source compaction' and 'target disambiguation' modules, alongside the use of multiple virtual prototypes, provides a robust foundation. These prototypes can potentially serve as initial cluster centers, allowing for further classification of unknown samples. While our current design doesn't directly address this, the foundational elements for such an extension are firmly in place.
>
> > **Q7. How are the experiments conducted with different types of baselines, including aspects like train-validation split and model selection? Further clarification and details regarding the fairness of comparison should be provided.**
>
> Our experiments, including train-validation splits and model selection, strictly adhere to the protocols established by previous OSDG works [70, 82]. We ensure a fair comparison by consistently applying these standards across different types of baselines. This guarantees that any observed performance differences stem from the methods themselves.

---

> > ### Comment · Reviewer_shJ8 · 2023-08-11
> > **Thanks for the response**
> >
> > The rebuttal solves all my concerns. After reading reviews from other reviewers and the responses, I do believe this work poses an interesting solution regarding a realistic problem, and it should be accepted. I upgrade my rating to "strong accept".

---

### Official Review · Reviewer_nkce · 2023-07-04

**Soundness:** 3 good
**Presentation:** 3 good
**Contribution:** 2 fair
**Rating:** 6
**Confidence:** 3

**Summary:**

This research paper introduces a novel experimental configuration referred to as Open Test-time Domain Generalization (OTDG), which can be viewed as an amalgamation of domain generalization, test-time adaptation, and open-set recognition. In order to address this particular experimental configuration, the authors put forward a fresh conceptual framework termed Compaction and Disambiguation (CODA). The fundamental concept behind this framework entails the imposition of restrictions on decision boundaries during the training phase, leveraging labeled source data, with the aim of ensuring an ample latent space to accommodate future classes. Subsequently, these decision boundaries are further refined during the test phase through the utilization of unlabeled target data.

**Strengths:**

1, The experimental setting introduced in this study represents a novel paradigm. Although its practical viability is yet to be established, it presents a unique problem that has not been previously encountered.

2, The paper effectively presents a well-structured argument, providing a reasonable proposition of loss functions aimed at addressing the OTDG problem.

3, The experiments conducted in this study are meticulously executed and accompanied by illustrative figures that effectively convey the findings. The evaluation involves a comprehensive comparison with recent state-of-the-art works that are closely related to the subject matter, ensuring a robust analysis of the proposed approach.

**Weaknesses:**

The motivation behind the concept of Open Test-time Domain Generalization (OTDG) in this study may benefit from further clarification. It would be valuable to present realistic scenarios that exemplify the challenges addressed by this setup. By providing concrete examples or case studies that demonstrate the practical relevance of OTDG, the study can establish a stronger connection between the proposed approach and its potential real-world applications. This would enhance the understanding of how OTDG can be applied in practical scenarios and highlight its significance in addressing specific challenges faced in domains that require generalization, test-time adaptation, and open-set recognition. The proposed solution in this study demonstrates a reasonable level of novelty in addressing the challenges posed by the specific OTDG problem. While it is acknowledged that there may be similarities with previous related works, this is considered a minor aspect. The study's main focus lies in presenting a solution that effectively tackles the challenges unique to the OTDG problem, highlighting the authors' contribution in the field. By emphasizing the novel aspects of the proposed solution and its potential impact, the study provides valuable insights and advancements within the context of the OTDG problem.

Minor weakness: no codes in the supplementary material.

**Questions:**

As I said above, can the authors list some practical scenario of the OTDG setup and convince me this is not a hand made setup just for research.

**Limitations:**

One of the main limitations of this study revolves around the practicality of the proposed setup. It is natural to question whether the setup accurately reflects real-world scenarios and if it can be applied in practical contexts. To gain more insight into the effectiveness and applicability of the Open Test-time Domain Generalization (OTDG) problem, it would be valuable to explore realistic scenarios that closely resemble the challenges faced in practical settings. This would help establish a stronger connection between the proposed approach and its potential real-world applications.

---

> ### Author Rebuttal · Authors · 2023-08-09
>
> Thank you for the constructive feedback and for acknowledging the innovative aspects of our study. We address the concerns as follows and will improve the manuscript accordingly.
>
> > **Q1. List some practical scenarios of the OTDG setup.**
>
> Great question! We would like to discuss three practical scenarios of OTDG:
>
> **(1)** Suppose we aim to integrate modern vision systems into autonomous vehicles. When shifts are limited to the environment (such as weather and illumination) or the appearance (such as size and viewpoint) of previously observed objects, current state-of-the-art techniques can correct for the potential shifts on the fly, such as Domain Adaptation (DA) and Domain Generalization (DG). But what if the sudden arrival of *new* objects in an *ever-changing* world? Most existing approaches will break and are likely to result in catastrophe, raising strong concerns about model safety and reliability. This brings two new challenges: 1) open and ever-changing environments hinder the usage of traditional DA (need access to target data for joint training with source domain) and DG (face significant performance degradation due to the open class and adaptivity gap) methods, and 2) test-time adaptation offers a promising direction but lacks principled solutions for addressing the open class in the context of DG. To this end, the proposed OTDG setup and CODA framework align precisely with this direction and provide compelling empirical results.
>
>  **(2)** When deploying a machine learning system tailored to diagnose diseases based on clinical symptoms, the underlying pre-assumption is that the data distribution remains stable over time. However, diseases don't always follow the IID and closed-set assumptions, given their ever-evolving nature. For example, the situation becomes dicey when we are blindsided by novel diseases, such as COVID-19 and its successive variants.
>
>  **(3)**  Let's further consider the application of machine learning systems in industrial anomaly detection. Typically, these systems are trained on historical data, assuming that machine operations and associated data patterns remain consistent. Using this data, techniques like Anomaly Detection (AD) are employed to identify and rectify abnormal behaviors. However, in a rapidly modernizing industrial landscape, novel machinery and processes are introduced frequently. What happens when an unprecedented anomaly, resulting from these new introductions, manifests? Traditional models might fail to detect or even misclassify such anomalies, potentially causing equipment damage or production halts.
>
> > **Q2. Minor weakness: no codes in the supplementary material.**
>
> Thank you for raising this point. We intend to release the code upon the paper's acceptance.

---

> > ### Comment · Reviewer_nkce · 2023-08-11
> > **Thanks for the response**
> >
> > Thanks for your response. I am happy to see that the authors are willing to include these discussions the in the future version or even the future supplementary material. Anyway, I am also willing to see the codes are released later to make people follow your job. In the end, I will still keep my positive attitude to accept this paper.

---

### Official Review · Reviewer_BmaN · 2023-07-05

**Soundness:** 4 excellent
**Presentation:** 4 excellent
**Contribution:** 3 good
**Rating:** 8
**Confidence:** 4

**Summary:**

This paper focuses on the problem of domain adaptation. In particular, in addition to the already-studied setting of domain shift at test time, this paper considers the practical scenario of unknown classes being presented at test time. Often current models will incorrectly predict a known class label for examples from unknown classes. This paper addresses this by: Compaction and Disambiguation. Compaction forces the known classes embeddings to be more compacted, with regions in between classes being reserved for unknown classes through the introduction of virtual unknown classes. Disambiguation, done at test time, is for test time adaptation. The objective in this phase optimizes 1) consistency between model output predictions using a nearest neighbor memory bank of class prototypes 2) consistency between source and target model to avoid catastrophic forgetting.

**Strengths:**

1) Introduction of a novel problem: unknown classes at test time in addition to domain shift

2) Thorough ablation on various components of methods to illustrate why every choice is necessary

3) Strong empirical results (SOTA across several datasets - I am not too familiar with what the standard datasets for domain generalization are though)


**Weaknesses:**

1) More background on test-time adaptation can be given before introducing the "disambiguation" protocol since many readers (like myself) may not be very familiar with this



**Questions:**

Can the authors explain in more detail why the known-class examples must activate unknown-class logits?

**Limitations:**

The authors could include a study on how the test time adaptation affects source domain accuracy i.e. what is it before and after the disambiguation protocol? I'm not sure if these results have been included or discussed thoroughly. Since the TTA loss has a term to encourage consistency with source model, how good the model is at this objective should be evaluated.

---

> ### Author Rebuttal · Authors · 2023-08-09
>
> Thanks very much for your insightful comments and suggestions! We have properly revised our paper based on your reviews and below are our detailed responses.
>
> > **Q1. More background on test-time adaptation can be given before introducing the "disambiguation" protocol.**
>
> We appreciate the reviewer's suggestion. Test-Time Adaptation (TTA) is an emerging paradigm that aims to adapt a source-trained model to novel target environments by progressively tuning the model weights using unlabeled test data during inference time. Compared to the traditional domain generalization pipeline, TTA mitigates the potential optimality gap (see Definition 1) and thus makes the whole process more practical for real-world applications. For example, suppose that we wished to deploy modern vision systems in an autonomous vehicle or a robot to recognize objects in the wild. Given that test environments can unpredictably encounter new objects at different stages, it is inefficient and even impossible to collect a set of labeled data to retrain the model. TTA exactly addresses this challenge, presenting a promising solution. Despite the promise, state-of-the-art TTA methods struggle to handle the open class as they typically seek the complete semantic alignment (i.e. cross-domain one-vs-one alignment) between source and target classes, resulting in the ''ambiguation'' between known and unknown classes (i.e. the source and target label spaces are asymmetric). We have integrated the aforementioned discussion into the revised manuscript for greater clarity.
>
> > **Q2. Why the known-class examples must activate unknown-class logits?**
>
> In our setting, we have no access to real unknown-class examples during the training phase. Consequently, we propose to use real known-class examples (second term in Eq. 3) and virtual unknown-class examples (first term in Eq. 4) to activate unknown-class logits. Empirically, as shown in Table 4, the prediction performance will significantly degenerate when only using Eq. 4. Here, we provide two perspectives to interpret why the known-class examples must activate unknown-class logits (second term in Eq. 3). **(1)** The response region of unknown classes (i.e., the gray regions in Fig. 2) is positively correlated to the size of unknown-class logits. Therefore, activating the unknown-class logits of known-class examples helps to enlarge the response region of unknown classes, thereby enhancing the unknown-aware ability of the source-trained model. **(2)** From the other perspective, since the virtual unknown classes are synthesized from known-class examples, activating the unknown-class logits of known-class examples will make the data manifold more smooth. As shown in Fig. 2, the virtual unknown-class samples (i.e., dashed points) compact the space of known-class samples (Fig. 2(b) vs. Fig. 2(c)), making the high-density regions of virtual unknown classes and known classes much closer and evenly distributed. This, in turn, smoothens the boundary delineations between known and unknown classes.
>
> > **Q3. The authors could include a study on how the test time adaptation affects source domain accuracy i.e. what is it before and after the disambiguation protocol?**
>
> Thank you for pointing out this crucial and often overlooked issue. We have added the results regarding how the TTA affects source domain accuracy in the newly attached PDF document. As depicted in Fig. 1 of the document, we can observe a noticeable decline in source domain accuracy as testing proceeds when the semantic consistency constraint is removed. In contrast, by ensuring semantic consistency between the target and original source models, our approach ensures robustness throughout the test-time adaptation process.

---

> > ### Comment · Reviewer_BmaN · 2023-08-10
> >
> > Thank you for your response. Including these revisions will strengthen the manuscript.
> >
> > I stand by my original recommendation to accept the paper.

---

### Official Review · Reviewer_bs2E · 2023-07-10

**Soundness:** 3 good
**Presentation:** 1 poor
**Contribution:** 3 good
**Rating:** 7
**Confidence:** 4

**Summary:**

This paper introduces a novel problem formulation termed Open Test-Time Domain Generalisation, which accounts for the presence of both domain shift and unseen classes during the testing phase. To tackle this problem, a two-stage methodology is proposed, comprising Compaction and Disambiguation. The primary objective of the Compaction stage is to facilitate the creation of a region in the embedding space that can accommodate the incoming unknown classes at test-time. This is accomplished by introducing synthetic unknown classes into the latent space by compacting the embedding space associated with the known classes from the source domain.  At training time, in lieu of unknown class data, manifold mixup is used to synthesise the unknown class training data and a classifier is trained that has additional decision boundaries corresponding to the unknown classes.

While the compaction stage is proposed during training to accommodate new classes, the disambiguation phase is proposed to handle the potential domain shift in the target domain. For the disambiguation phase, authors adopt existing TAST [ref 20 from paper] methodology and apply it to a setting with unknown classes.

Experimental results show that  the current method has been exhaustively tested against baselines.

**Strengths:**

+ Novel Problem Formulation: The paper introduces a novel problem formulation called Open Test-Time Domain Generalization, which addresses the challenges of domain shift and unseen classes during the testing phase. This formulation extends the scope of traditional domain generalization methods and accounts for realistic scenarios where new classes can emerge at test time.
+ Integration of Manifold Mixup: To address the lack of unknown class data during training, the paper leverages manifold mixup to synthesize the unknown class training data. This integration allows the model to learn decision boundaries corresponding to the unknown classes. By utilizing this data augmentation technique, the proposed method enhances the model's ability to generalize to unknown classes and improve overall performance.
+ Extensive Experimental Evaluation: The paper demonstrates the thoroughness of the research by subjecting the proposed method to exhaustive testing against relevant baselines.


**Weaknesses:**

The work brings together ideas from different existing methods and provides a novel way to tackle the problem setting of Open Test-Time Domain Generalisation. While there are a certain amount of novel claims made in the paper, they are often unsubstantiated. We have provided some pointers to where further explanations are required. Additionally, we recommend that paper be thoroughly proofread.

* The paper lacked a sufficient level of explicitness in conveying the necessary information. Important details or background knowledge seemed to be assumed or left to inference, which makes it challenging for readers to follow the logical flow of the study. It would greatly benefit the clarity of this work if additional context, explanations, and descriptions could be provided to ensure a more straightforward understanding for readers who may not possess the same background knowledge or expertise as the authors. For instance, (a)
While Eq.5 shows that the prototype-based classifier assigns 'z' to one of |C_s|+N classes, line 60 of the supplementary file contradicts that by assuming all N classes to be one class. Please clarify. (b) The mechanics of support set, its constitution and updation are ill-defined.
* While the paper asserts the mitigation of catastrophic forgetting in line 16, it lacks empirical evidence to validate this claim. Therefore, it is recommended to include relevant results in order to substantiate the effectiveness of the proposed approach in addressing catastrophic forgetting. Reporting ablation study of semantic consistency loss on source domain known classes is advised.
* In line 133, the inequality seems to be incorrect and the assertion in lines 134-135 does not trivially follow from the inequality.
* Lines 132 through 136 introduce the notion of optimality gap which we believe is a misnomer as this is defined as “adaptivity gap” [ref 11 from paper]. However, regardless of the appropriateness of the term optimality gap, we believe that the problem that TTA resolves is domain shift, rendering the previous lines irrelevant.


**Questions:**

* While the statement “...we activate the response of the real known class towards unknowns (Eq. 3)” from line 124 is adequately explained, we believe that further explanation of line 125, “... and the response of the virtual unknown class towards the known ones” is required.
* In line 111, y_hat shouldn’t be the index but rather a one-hot vector with non-zero entry at the index that y_hat currently holds. Similarly with line 121, “ y’ ” is incorrectly defined.
* In Eq.3 and Eq.4, a new undefined term ‘C’ is introduced.
* We request the authors to use notation consistent with TAST [ref 20 from paper] when building upon their contributions, for increased accessibility.
* Please justify the choice of scoring function in line 165.
* In line 185, the unknown class set cannot include the source domain class set.
* Line 205 requires clarification.
* While Eq.5 shows that the prototype-based classifier assigns 'z' to one of |C_s|+N classes, line 60 of the supplementary file contradicts that by assuming all N classes to be one class. Please clarify.


**Limitations:**

None in particular

---

> ### Author Rebuttal · Authors · 2023-08-09
>
> We sincerely appreciate the reviewer for the positive feedback and insightful comments! We extend our apologies for any inconvenience or misunderstanding that may have occurred when reading our manuscript, and we assure you that we will incorporate all of your valuable suggestions in our final version. Our responses to your specific questions are listed below.
>
> > **Q1. While Eq.5 shows that the prototype-based classifier assigns 'z' to one of |C_s|+N classes, line 60 of the supplementary file contradicts that by assuming all N classes to be one class.**
>
> We apologize for the confusion. The statement in line 60 of the supplementary file is indeed imprecise. What we intended to convey is that our method aims to classify unknown classes into various virtual unknown classes. These can essentially be regarded as a whole, as we did not pursue the precise clustering of the unknown classes themselves. We will deliver this point more precisely in the revised manuscript.
>
> > **Q2. The mechanics of support set, its constitution and updation are ill-defined.**
>
> The constitution and updation of the support set (aka. memory bank) were defined in lines 141-155 (main text). To facilitate a better understanding, we highlight several key aspects: (1) the support set consists of two types of sources: penultimate features and logits; (2) for each test sample, we compute its nearest neighbor from the support set; (3) the local class conditionals are estimated from the nearest neighbor set (Eq. 6); (4) the estimated local class conditionals are used to update the global class prototype of the support set (Eq. 7) in a moving-average style.
>
> > **Q3. The effectiveness of the proposed approach in addressing catastrophic forgetting. Reporting ablation study of semantic consistency loss on source domain known classes is advised.**
>
> Nice question! As suggested, we report the ablation of semantic consistency loss on source domain known classes in the newly attached PDF document. As depicted in Fig. 1 of the document, we can observe a noticeable decline in source domain accuracy as testing proceeds when the semantic consistency constraint is removed. In contrast, by ensuring semantic consistency between the target and original source models, our approach ensures robustness throughout the test-time adaptation process.
>
> > **Q4. In line 133, the inequality seems to be incorrect and the assertion in lines 134-135 does not trivially follow from the inequality.**
>
> Correction: $\varepsilon_S(h^t)<\varepsilon_T(h^*)$ --> $\varepsilon_T(h^t)<\varepsilon_T(h^*)$. We are sorry for the lack of carefulness. This inequality represents that the target optimal classifier $h^t$ will perform better than the universal optimal classifier $h^*$ on the target domain $T$. Consequently, in lines 134-135, we conclude that ''it is not feasible to find a universal optimal classifier that applies to both source and target domains'', highlighting the importance of our test-time adaptation module.
>
> > **Q5. Lines 132 through 136 introduce the notion of optimality gap which we believe is a misnomer as this is defined as “adaptivity gap” [ref 11 from paper].**
>
> The authors of [11] conceptualized ''adaptivity gap'' in the context of DG but did not provide a detailed mathematical definition. We formalize this issue and rename it as ''optimality gap'' to reflect its inner characteristics. To keep the consistency, we will adhere to the naming used in [11] (i.e. adaptivity gap) in the final version.
>
> > **Q6. Further explanation of line 125, “... and the response of the virtual unknown class towards the known ones” is required.**
>
> As shown in Figure 1, Eq. (3) and Eq. (4) are symmetric. In Eq. (3), real known-class samples will give the largest response to its ground-truth (known-class) dimension, and the second-largest response to a certain unknown-class dimension. Similarly, In Eq. (4), virtual unknown-class samples will give the largest response to its ground-truth (unknown-class) dimension (first term), and the second-largest response to a certain known-class dimension (second term).
>
> > **Q7. In line 111, y_hat shouldn’t be the index but rather a one-hot vector with non-zero entry at the index that y_hat currently holds. Similarly with line 121, “ y’ ” is incorrectly defined.**
>
> We apologize for this oversight. You are absolutely correct. To align with Eq. (1) and Eq. (2), we revise the second term of Eq. (3) and Eq. (4) as:
>
> * $-\log \frac{ \exp (f_{\hat{y}}(\mathbf{x}))}{\sum_{c\in |\mathcal{C}_s|+N, c \neq y} \exp (f_c(\mathbf{x}))}$
> * $-\log \frac{ \exp (h_{y'}(\hat{\mathbf{z}}))}{\sum_{c\in |\mathcal{C}_s|+N, c \neq \hat{y}} \exp (h_c(\hat{\mathbf{z}}))}$
>
> > **Q8. In Eq.3 and Eq.4, a new undefined term ‘C’ is introduced.**
>
> Sorry for the typo. The 'C' here corresponds to 'N', i.e.,  the number of virtual class prototypes defined in line 108.
>
> > **Q9. We request the authors to use notation consistent with TAST [ref 20 from paper] when building upon their contributions, for increased accessibility.**
>
> Thank you for your suggestion. We will revise our manuscript to align notations with [20].
>
> > **Q10. Please justify the choice of scoring function in line 165.**
>
> Eq. (10) is designed to exclude unreliable samples as determined by their entropy during test-time adaptation, and it gives more weight to reliable samples that have lower prediction uncertainties. By doing so, we ensure that those reliable samples exert a greater influence during the test-time model updating.
>
> > **Q11. In line 185, the unknown class set cannot include the source domain class set.**
>
> Sorry for the typo. 'combined with $\mathcal{C}_s$' should be removed.
>
> > **Q12. Line 205 requires clarification.**
>
> Following [82], a target sample is marked as an unknown class if its entropy is larger than a threshold, which is empirically set to $\log(|\mathcal{C}_s|)/2$ where $|\mathcal{C}_s|$ represents source label space and $\log(|\mathcal{C}_s|)$ is the maximum value of entropy.

---

> > ### Comment · Reviewer_bs2E · 2023-08-15
> >
> > Thank you to the authors for the responses. My concerns have been satisfactorily addressed. I believe the problem setting is novel but the components devised to make the contribution are all pre-existing. The novelty lies in bringing disparate components together in a fairly meaningful way (Mixup in compaction stage, semantic consistency and entropy based sample selection in disambiguation). I am increasing the score to Accept.
> >
> > My only pending (relatively minor) concern would be the presentation issues. Although the authors have acknowledged and agreed to fix all the editorial issues, the paper's presentation may not be fully fixed by proof reading and some deeper changes have to introduced into the manuscript for the paper to be accessible.

---

### Author Rebuttal · Authors · 2023-08-09

We thank all the reviewers for their time, insightful suggestions, and valuable comments. We are glad that **ALL** reviewers appreciate our work and find our method simple and effective, highlighting that a novel problem (bs2E, BmaN, nkce, shJ8) - Open Test-Time Domain Generalization (OTDG), a novel and technically sound framework (shJ8) - CODA, extensive experimental evaluation (bs2E, BmaN, nkce, shJ8), strong empirical results (BmaN), and thorough ablation (BmaN, shJ8). We are also encouraged that reviewers find the paper is well organized (shJ8) and presents a well-structured argument (nkce).

We respond to each reviewer's comments in detail below. We have incorporated the reviewers' suggestions into our manuscript, which we believe has significantly strengthened the paper. The main changes we made include:

* We report the source domain accuracy w/ and w/o semantic consistency loss (as Reviewer bs2E and Reviewer BmaN suggested).
* We provide clarifications and corrections to address the reviewer's questions about the main text and have rigorously proofread the entire manuscript (as Reviewer bs2E suggested).
* We provide more background regarding TTA and clarify the motivation for using the known-class examples to activate unknown-class logits (as Reviewer BmaN suggested).
* We list some practical scenarios of the OTDG setup (as Reviewer nkce suggested).
* We offer detailed explanations and clarifications on certain technical aspects (as Reviewer shJ8 suggested).

---

### Decision · Program_Chairs · 2023-09-21

**Decision:**

Accept (spotlight)

**Comment:**

The paper introduces Open Test-Time Domain Generalization (OTDG), a new concept to handle both domain shifts and unseen classes during testing. The approach has two stages: Compaction, which uses manifold mixup to prepare for unknown classes, and Disambiguation, which deals with domain shifts. Initially, the major concern is that it lacks clarity in some areas and could benefit from further validation and context.

The reviewers are all satisfied with the authors' response and recommend accept. AC reads the paper (roughly), the review, and the discussion, and recommends accept (spotlight).